# Phytoplankton Structure and Ecological Niche Differentiation of Dominant Species in Tahe Bay, China

**DOI:** 10.3390/biology14050578

**Published:** 2025-05-21

**Authors:** Yanrong Zhang, Zengqiang Yin, Yinghai Du, Xiangxu Wei, Yikai Lan, Quan Yu, Yan Wang, Tao Tian, Lei Chen, Jun Yang

**Affiliations:** 1College of Marine Science and Environment, Dalian Ocean University, Dalian 116023, China; 18863092987@163.com (Y.Z.); d378840134@163.com (Y.D.); 18840870087@163.com (X.W.); lanyikai2002@163.com (Y.L.); wangy@dlou.edu.cn (Y.W.); chenlei@dlou.edu.cn (L.C.); yangj@dlou.edu.cn (J.Y.); 2College of Fisheries and Life Science, Dalian Ocean University, Dalian 116023, China; 18957231585@163.com (Q.Y.); tian2007@dlou.edu.cn (T.T.)

**Keywords:** phytoplankton structure, ecological niche width, ecological niche overlap, environmental factors, redundancy analysis

## Abstract

Phytoplankton are the basis of the productivity of the sea area, and understanding the characteristics of the phytoplankton community is a guarantee for the sustainable development of local marine fisheries. A survey was conducted in the waters of Tahe Bay to study the characteristics of the phytoplankton structure and the dominant species’ niche. The results showed that 83 phytoplankton species were identified in six phyla and forty-one genera, dominated by Bacillariophyta, and the phytoplankton community has high species diversity and strong resistance to external disturbances. The dominant species, 20 species in three phyla, mainly consisted of wide-niche species; *Paralia sulcata* was always the dominant species except in summer. The niche overlap of dominant species was highest in winter, followed by spring, autumn, and summer. The abundance and community diversity of phytoplankton were significantly correlated with environmental variables; an increase in NO_3_-N and DIP would probably inhibit the growth of species in the Pyrrophyta, Chrysophyta, and Cyanophyta. COD, water temperature, NO_3_-N, DIP, NO_2_-N, and NH_3_-N were the main environmental factors affecting the ecological niche differentiation of dominant phytoplankton species. The results of the study provide scientific references for a deeper understanding of the sustainability of local ecosystems and thus promote high-quality mariculture development.

## 1. Introduction

Phytoplankton are primary producers in marine ecosystems, distinguished by short life cycles, accelerated reproduction rates, and swift responses to environmental variables [1,2]. They can influence the productivity of marine economic organisms directly or indirectly through the trophic cascade effect within the food chain [3]. Consequently, investigating the spatial and temporal variations in phytoplankton structure might offer a theoretical foundation for enhancing marine fisheries production. The dominant species are the main components of the community, and their ecological niche attributes reflect the structural characteristics, function, and succession of the community and the evolution of the population [4,5]. Hence, examining phytoplankton structure and the ecological niches of dominant species is crucial for comprehending the stability of biological communities, the extent of resource utilization and competition among dominant species, and investigating the interactions between phytoplankton and their environment, which can inform the restoration and management of marine habitats [6].

Phytoplankton surveys in the waters of Dalian have been conducted early on; for instance, Zhao et al. [7] examined the species composition, density, and diversity of phytoplankton in Dayaowan Bay based on data collected in 2010. Jiang et al. Ref. [8] elucidated the ecological characteristics of the phytoplankton community in the red tide monitoring area of Swertia Island, Dalian. Song et al. Ref. [9] analyzed the interannual variations in phytoplankton structure in the eastern Dalian aquaculture area during the summer months from 2012 to 2016. Chen Ying et al. Ref. [1] investigated the seasonal variations in phytoplankton and the environmental factors influencing them in the offshore aquaculture area of Dalian, utilizing annual data from January to October 2020. Nevertheless, the research mentioned above mostly concentrated on the structure of phytoplankton communities and their environmental correlations, and no published reports have been published on the investigation of phytoplankton ecological niche differentiation in the waters of Dalian.

Lushun Tahe Bay is situated in the northern waters of the Yellow Sea, which is the confluence area of the warm currents of the Yellow Sea and the coastal currents of Liaodong, and is rich in biological resources, establishing it as a historically famous fishing ground. Due to the serious decline of fishery resources at the end of the 20th century, the local government has carried out large-scale construction of artificial reefs and algal floating rafts to restore fishery resources and improve the quality and production of aquatic products, which is one of the most important aquaculture areas in the northern part of the Yellow Sea. The artificial reefs and algal rafts together form a typical composite ecosystem in this area. With an area of about 7 km^2^, a water depth of around 10 m, a water temperature of 5~25 °C, and a salinity of 30.2~32.2, and influenced by the warm current of the Yellow Sea and the coastal current, the nutrient salts are abundant, and seawater exchanges are adequate [10,11]. The major species in the waters of Tahe Bay include *Undaria pinnatifida Suringar*, scallops, sea cucumbers, and coral reef fishes [12]. The construction of artificial reefs and the installation of aquaculture rafts have altered the local flow distribution, optimizing nutrient salt composition and impacting the phytoplankton community [13,14,15]. This study analyzed the phytoplankton structure and the ecological niches of dominant species (width and degree of overlap) in Tahe Bay, utilizing survey data from four seasons between September 2021 and November 2022. By integrating environmental characteristics, the research explored the ecological niche differentiation of dominant species, contributing to an understanding of the dynamics of the phytoplankton community. This will elucidate the alteration rules of dominant species and the stability of the community structure of the phytoplankton community in this sea area, thereby offering a theoretical foundation for the scientific assessment of the ecological health of the area and the sustainable utilization of marine biological resources.

## 2. Materials and Methods

### 2.1. Survey Time and Stations

A fishing vessel was chartered for sampling in Tahe Bay, conducting studies of the seawater environment and phytoplankton in September 2021 (late summer), mid-March 2022 (late winter), late April 2022 (spring), and November 2022 (autumn). (Since sampling was limited by the annual fishing closed season in May–August, the samples were collected in late April for spring and late September for summer, based on the water temperature at the time of the surveys, thereby the survey samples included essentially all seasons in these waters) [11]. Figure 1 illustrates the distribution of survey stations.

### 2.2. Sample Collection and Processing

Qualitative samples of phytoplankton were obtained using a shallow water type III phytoplankton net (inner diameter of 37 cm, aperture of 0.076 mm, net length of 140 cm, and net area of 0.10 m^2^) at each station by vertically towing the net from the bottom to the water surface, and the filtrate was treated and kept in 5% formaldehyde [16]. For quantitative samples, 1 L of surface and bottom water samples were collected using a water harvester, placed in polyethylene bottles, fixed with 15 mL of Lugol’s solution, and subsequently concentrated to 50 mL after settling in the laboratory. Samples were subsequently aspirated and subjected to qualitative and quantitative analysis with an Olympus X31 biomicroscope.

### 2.3. Measurement of Physical and Chemical Indicators of Water Quality

The pH, water temperature (T), salinity (SAL), and dissolved oxygen (DO) of the water body at the survey stations were measured on-site using the MultiAnna MTA 5 multiparameter water quality analyzer. Concurrently, additional water samples were transported to the laboratory, where the chemical oxygen demand (COD, mg/L) was assessed using the potassium dichromate method; ammonia nitrogen content (NH_3_-N, mg/L) was measured via Nano reagent spectrophotometry; nitrate nitrogen content (NO_3_-N, mg/L) was measured through phenol disulphonic acid spectrophotometry; nitrite nitrogen content (NO_2_-N, mg/L) was evaluated by the spectrophotometric analysis; dissolved inorganic phosphorus (DIP, mg/L) was quantified using ammonium molybdate spectrophotometry; and the sulfide content (S, mg/L) was analyzed with methylene blue spectrophotometry. The specific determination technique refers to [17].

### 2.4. Data Analysis and Processing

#### 2.4.1. Degree of Dominance Index

The degree of dominance, which was calculated using the following formula, was employed to identify the dominant phytoplankton species [18,19]:(1)$$Y=\frac{N_{i}}{N}\times F_{i}$$
where *N* represents the total number of individuals in all species; *N_i_* represents the number of species *i*; and *F_i_* represents the frequency of occurrence of species *i*. In accordance with a related study [20], the species is dominant when *Y* is greater than or equal to 0.02.

#### 2.4.2. Biodiversity Indicators

This article utilized Shannon–Wiener’s diversity index (*H′*) [21], the Pielou index (*J*) [22], and Margalef’s index (*D*) [23,24,25] to assess the diversity of phytoplankton communities in Tahe Bay’s waters. The diversity of the phytoplankton community was assessed using the following formulas:(2)$$H\prime =-\sum P_{i}\ln P_{i}$$(3)$$J=\frac{H\prime }{\ln S}$$
(4)$$D=\frac{\left(S-1\right)}{\ln N}$$

*P_i_* represents the ratio of the number of individuals of the *i*-th species in the community to the total number of individuals of the species in the formulae; *S* denotes the number of species in the community; and *N* signifies the aggregate number of individuals of all species.

#### 2.4.3. Ecological Niche Width Index

The niche width index serves as a crucial metric for evaluating the adaptation of organisms to their environment and their capacity to exploit diverse resources [26]. This research uses the Levin’s formula to compute the niche width of dominating species [27]. The formula is as follows:(5)$$B_{i}=\frac{1}{r\sum_{j=1}^{1}p_{ij}^{2}}$$

*P_ij_* represents the proportion of individuals of species *i* that utilize resource *j*; whereas *r* denotes the total number of resources, which in this case corresponds to the total number of sampling stations. A higher *B_i_* value corresponds to a broader biological niche width for the species, and a similar study [28] indicates that *B_i_* ranges from 0 to 1. The species is classified as a wide ecological niche when *B_i_* ≥ 0.6, a medium ecological niche when 0.3 ≤ *B_i_* < 0.6, and a restricted ecological niche when *B_i_* < 0.3.

#### 2.4.4. Pianka Ecological Niche Overlap Index

The ecological niche overlap index is a quantitative metric employed in ecology to assess the extent of similarity among various species regarding resource utilization, spatial occupation, and requisite environmental conditions [29]. The Pianka overlap index was employed to quantify the extent of ecological niche overlap utilizing the subsequent formula [30]:(6)$$Q_{ik}=\frac{\sum_{j=ij}^{N}\cdot p_{kj}}{\sqrt{\sum_{j=1}^{N}p_{ij}^{2}\cdot \sum_{j=1}^{N}p_{kj}^{2}}}$$

*N* represents the total number of stations within the surveyed area, while *P_ij_* and *P_kj_* denote the proportions of phytoplankton *i* and phytoplankton *k*, respectively, in the overall phytoplankton population at station *j*. The quantities of phytoplankton *i* and *k* in the examined region correspond to the aggregate count of phytoplankton at station *j*. A related study [28] indicates that the niche overlap of species *i* and *k* is low when the niche overlap degree *Q_ik_* is less than 0.3, meaningful when *Q_ik_* is between 0.3 and 0.6, and substantial when *Q_ik_* is greater than or equal to 0.6.

#### 2.4.5. Statistics

ArcMap 10.7 was employed to delineate the survey stations; SPSS 22.0 facilitated Pearson’s correlation analysis on phytoplankton diversity indices (*H′*, *J*, *D*), abundance, and environmental factors. Environmental factors exhibiting a variance inflation factor (VIF) exceeding 10 were manually excluded to ascertain environmental factors significantly influencing the structure of the phytoplankton community and the correlation degree between phytoplankton diversity indices and environmental factors. Canoco 5.0 was employed to ascertain the length of the primary correlation axis between the dominant species of phytoplankton and environmental data through detrended correspondence analysis (DCA). This analysis determined the appropriate method for subsequent analysis: if the first axis value of DCA exceeded 4.0, canonical correspondence analysis (CCA) was conducted; if the value ranged from 3.0 to 4.0, either redundancy analysis (RDA) or CCA could be utilized; if the value was below 3.0, RDA was favored. The bar charts were generated using Excel 2021.

## 3. Results and Analyses

### 3.1. Analysis of the Phytoplankton Structure

#### 3.1.1. Phytoplankton Species Composition and Species Diversity

Figure 2 illustrates the fluctuations in the abundance of diverse species in the waters of Tahe Bay from September 2021 to November 2022. The abundance of phytoplankton in Tahe Bay ranged from 14.90 × 10^3^ to 17.46 × 10^5^ ind·m^−3^ across all seasons, with a mean abundance of 22.13 × 10^4^ ind·m^−3^. The seasons in descending order in terms of the total number of phytoplankton species were winter, spring, autumn, and summer. The Bacillariophyta had the greatest mean abundance, followed by Cyanophyta. In March 2022, Bacillariophyta exhibited the highest abundance at 17.46 × 10^5^ ind·m^−3^; the other phyla reached their peak abundance in November 2022, with Cyanophyta at 11.16 × 10^3^ ind·m^−3^, Pyrrophyta at 5500 ind·m^−3^, Chlorophyta at 18.70 × 10^2^ ind·m^−3^, Chrysophyta at 11.20 × 10^2^ ind·m^−3^, and Rhodophyta at 510 ind·m^−3^.

Between September 2021 and November 2022, four surveys found a total of 83 phytoplankton species in six phyla and forty-one genera, including Bacillariophyta, Cyanophyta, Pyrrophyta, Chlorophyta, Chrysophyta, and Rhodophyta. The Bacillariophyta comprised the largest number of species, totaling 65, which represented 78.31%; the Pyrrophyta included 8 species, accounting for 9.64%; the Cyanophyta had 4 species, constituting 4.82%; the Chlorophyta contained 3 species, making up 3.61%; the Rhodophyta featured 2 species, corresponding to 2.41%; and the Chrysophyta encompassed 1 species, equating to 1.20% (Table 1). The total number of phytoplankton species was highest in autumn and lowest in late summer, and the species composition of the Bacillariophyta was absolutely dominant in all seasons. Based on Equations (2)–(4), the phytoplankton diversity index ranged from 2.03 to 2.80, with a medium of 2.32, the richness index ranged from 0.83 to 4.99, with a median of 2.13, and the evenness index ranged from 0.28 to 0.84, with a median of 0.33. The peak values of *H’* and *D* were recorded in November 2022 (autumn), and the values of *H’* (2.80), D (4.99), and *J* were the highest in September 2021 (late summer) (0.84) and the lowest in November 2022 (late winter) (0.28) (Table 2).

#### 3.1.2. Analysis of Dominant Species

The study revealed that a total of 20 dominant species, belonging to three major phyla, were recorded during the four sampling periods. The phytoplankton ecotypes exhibited the dominance of wide-temperature and wide-salt species as well as temperate near-shore species, with Bacillariophyta being the most dominant species group in this sea area (Table 3). The dominant species included seventeen species of Bacillariophyta, two species of Pyrrophyta, and one species of Cyanophyta. The dominant phytoplankton species in Tahe Bay’s waters were as follows: In September 2021 (late summer), the dominant species comprised five: *Leptocylindrus danicus*, *Thalassiothrix longissima*, *Noctiluca scintillans*, *Trieres chinensis*, and *Coscinodiscopsis jonesiana*. In March 2022 (late winter), the dominant species increased to seven: *Paralia sulcata*, *Thalassiosira nordenskioeldii*, *Stephanopyxis turris*, *Chaetoceros densus*, *Chaetoceros peruvianus*, *Thalassionema nitzschioides*, and *Navicula cancellata*. By April 2022 (spring), the dominant species expanded to eight: *Paralia sulcata*, *Thalassiosira nordenskioeldii*, *Stephanopyxis turris*, *Chaetoceros densus*, *Chaetoceros peruvianus*, *Thalassionema nitzschioides*, *Navicula cancellata*, and *Skeletonema costatum*. In November 2022 (autumn), the dominant species totaled nine: *Leptocylindrus danicus*, *Paralia sulcata*, *Bacillaria paradoxa Gmelin*, *Coscinodiscus asteromphalus*, *Coscinodiscus granii*, *Licmophora abbreviata*, and *Melosira granulata var. angustissima*, *Synedra acus*, and *Tripos muelleri*. *Leptocylindrus danicus* was the first dominant species in the late summer (0.116), and *Paralia sulcata* was the first dominant species in the late winter, spring, and autumn (0.364, 0.391, and 0.165, respectively).

### 3.2. Ecological Niche Width (B_i_) and Overlap (Q_ik_) of Dominant Species

The niche widths and overlap degrees of dominant species in the four surveys conducted from September 2021 to November 2022 were computed using Equations (5) and (6) (Table 4, Table 5, Table 6 and Table 7). The niche width of dominant phytoplankton species in Tahe Bay varied from 0.44 to 1.00, and the overlap degree *Q_ik_* ranged from 0 to 1.00. The niche width of dominant phytoplankton species in Tahe Bay during September 2021 varied from 0.79 to 1.00 (Table 4), categorizing all as wide-niche species since *B_i_* > 0.6. The niche widths of *Trieres chinensis* and *Coscinodiscopsis jonesiana* were the largest, with a value of 1.00, followed by *Noctiluca scintillans* with a niche width of 0.95, and the smallest niche width of 0.79 was seen in *Thalassiothrix longissima*. The niche overlap of the dominant species was calculated using Pianka’s formula, revealing that phytoplankton niche overlap ranged from 0 to 1.00 in September 2021 (Table 4). There were four pairs with a substantial niche overlap (*Q_ik_* > 0.6), comprising 40%, with *Trieres chinensis* and *Coscinodiscopsis jonesiana* demonstrating a complete ecological niche overlap (1.00). There were six pairs with an ecological niche overlap index of less than 0.3, representing 60%, among which the ecological niche overlap index between *Leptocylindrus danicus* and *Trieres chinensis*, as well as between *Coscinodiscopsis jonesiana* and *Leptocylindrus danicus*, and *Noctiluca scintillans* species pairs was zero. The computation results are presented in Table 4.

The niche widths of the dominant species in March 2022 varied from 0.69 to 1.00 (Table 5), categorizing them all as wide-niche species due to *B_i_* > 0.6. The niche width of *Thalassionema nitzschioides* reached a maximum of 1.00, while the smallest niche width was only 0.69 for *Chaetoceros densus*. Utilizing Equation (6), it was found that the niche overlap index of dominant phytoplankton species in March 2022 ranged from 0.70 to 1.00 (Table 5). The number of species pairs with a substantial niche overlap (*Q_ik_* > 0.6) was twenty-one pairs, or 100% of the total, indicating a high degree of interspecific competition within the community.

In April 2022, the niche widths of the dominant phytoplankton species varied from 0.74 to 1.00 (Table 6), and all of them were wide-niche species since *B_i_* > 0.6. The niche width of *Thalassionema nitzschioides* is 1.00, whereas the niche width of only 0.74 for *Chaetoceros densus*, which has the smallest niche width. Utilizing Equation (6), the niche overlap index of dominant phytoplankton species in April 2022 varied between 0.45 and 0.99 (Table 6). There were twenty-six pairs with a substantial niche overlap (*Q_ik_* ≥ 0.6), accounting for 93%, and two pairs had a meaningful niche overlap (0.3 ≤ *Q_ik_* < 0.6), representing 7%.

In November 2022, the niche widths of the dominant phytoplankton species varied from 0.44 to 0.89 (Table 7), encompassing six broad niches and three medium niches. The maximum niche width value of 0.89 was observed in *Melosira granulata var. angustissima* followed by *Bacillaria paradoxa Gmelin*, which had a niche width of 0.79, while *Coscinodiscus granii* demonstrated the smallest niche width of just 0.44. Utilizing Equation (6), the phytoplankton niche overlap index varied from 0.17 to 1.00 in November 2022 (Table 7). Twenty-one pairs had a substantial niche overlap (*Q_ik_* > 0.6), or 58% of the total. The niche overlap index between *Leptocylindrus danicus* and *Synedra acus* was 1.00, indicating a high degree of niche overlap. Thirteen pairs exhibited a meaningful niche overlap (0.3 ≤ *Q_ik_* < 0.6), representing 36% of the total, and the niche overlap index between *Coscinodiscus granii* and *Synedra acus* was low, with a value of 0.31. There were two pairs with an ecological niche overlap index less than 0.3.

### 3.3. Phytoplankton Community Structure About Environmental Factors

Table 8 indicates that diversity index (*H’*) and richness (*D*) exhibited a highly significant positive correlation with salinity (*p* < 0.01) and a highly significant negative correlation with water temperature, nitrate nitrogen content, nitrite nitrogen content, and ammonia nitrogen content, respectively (*p* < 0.01); the evenness (*J*) demonstrated a completely contrary relationship to *H’* and *D*. The abundance of Bacillariophyta exhibited a highly significant positive correlation with dissolved oxygen concentration and inorganic phosphorus content (*p* < 0.01), along with a significant positive correlation with pH (*p* < 0.05), and the abundance of Pyrrophyta showed a significant negative correlation with nitrate nitrogen (*p* < 0.05) and a highly significant negative correlation with pH and inorganic phosphorus content (*p* < 0.01). The abundance of Chrysophyta and Cyanophyta revealed a highly significant negative correlation with pH (*p* < 0.01).

### 3.4. Factors Influencing Ecological Niche Differentiation of Dominant Phytoplankton Species

This paper first employed Pearson’s correlation analysis to examine the abundance of dominant phytoplankton species with environmental factors, excluding highly autocorrelated variables. Subsequently, the dominant species abundance data of each season in the waters of Tahe Bay were analyzed using DCA. The findings indicated that the longest gradient length of the first ordination axis for dominant phytoplankton species was greater than 3 in September 2021 (3.37), while it was less than 3 in March 2022 (0.07), April (0.60), and November 2022 (0.68). Consequently, the correlation between phytoplankton communities and environmental factors was analyzed and evaluated using the CCA method in September 2021 and the RDA method for the other three sampling periods.

Figure 3 illustrates that COD, NO_3_-N, NO_2_-N, DIP, and salinity had a greater impact on the phytoplankton community structure in Tahe Bay, Lushun, during September 2021. Among these, the environmental factors excluding DIP, NO_2_-N, and COD exhibited a negative correlation with the scores of the first principal component axis, whereas NH_3_-N, pH, salinity, DO, and NO_3_-N demonstrated a positive correlation with the scores of the second principal component axis. The abundance of the dominant species, *Leptocylindrus danicus* and *Noctiluca scintillans*, exhibited a positive correlation with COD, DIP, and NO_2_-N, while showing a negative correlation with water temperature, pH, NO_3_-N, salinity, and DO. Conversely, the abundance of *Thalassiothrix longissima*, *Trieres chinensis*, and *Coscinodiscopsis jonesiana* was positively correlated with water temperature, NO_3_-N, salinity, DO, and pH and negatively correlated with COD, DIP, and NO_2_-N.

Figure 4 illustrates that NH_3_-N, NO_3_-N, DIP, salinity, and water temperature had a greater influence on the phytoplankton community structure in Tahe Bay, Lushun, during March 2021. All environmental factors, except DIP, pH, NO_2_-N, and water temperature, exhibited a negative correlation with the scores of the first principal component axis, whereas NH_3_-N, salinity, DO, and COD demonstrated a positive correlation with the scores of the second principal component axis. The abundance of the dominant species, including *Stephanopyxis turris*, *Chaetoceros densus*, *Chaetoceros peruvianus*, and *Thalassiosira nordenskioeldii*, exhibited a positive correlation with COD and DO, while demonstrating a negative correlation with water temperature, pH, NO_2_-N, and DIP. Conversely, the abundance of *Paralia sulcata* and *Thalassionema nitzschioides* showed a positive correlation with water temperature, salinity, pH, NH_3_-N, and NO_2_-N and a negative correlation with DIP and NO_3_-N. Additionally, the abundance of *Navicula cancellata* was positively correlated with pH, NO_3_-N, NO_2_-N, and DIP.

Figure 5 illustrates that COD, NH_3_-N, NO_3_-N, NO_2_-N, DIP, and water temperature had a greater influence on the phytoplankton community structure in Lushun, Tahe Bay, in April 2022. Notably, all environmental factors, except salinity, NO_3_-N, pH, COD, and NO_2_-N, exhibited a negative correlation with the scores of the first principal component axis; conversely, salinity and pH were negatively correlated with the scores of the second principal component axis. The abundance of the dominant species, *Paralia sulcata*, *Stephanopyxis turris*, *Chaetoceros densus*, *Chaetoceros peruvianus*, and *Thalassiosira nordenskioeldii*, exhibited a positive correlation with DO, DIP, and NH_3_-N, while showing a negative correlation with salinity and pH. Conversely, the abundance of *Thalassionema nitzschioides*, *Navicula cancellata*, and *Skeletonema costatum* was positively correlated with water temperature, salinity, pH, NO_3_-N, and NO_2_-N and negatively correlated with DO.

Figure 6 illustrates that DIP, NO_2_-N, DO, COD, and water temperature had a greater influence on the phytoplankton community structure in Tahe Bay, Lushun, in November 2022. Water temperature, salinity, and pH exhibited a positive correlation with the scores on the first principal component axis, while all environmental factors, except water temperature, salinity, pH, DO, and NH_3_-N, were negatively correlated with the scores on the second principal component axis. The abundance of *Bacillaria paradoxa Gmelin*, *Licmophora abbreviata*, *Melosira granulata var. angustissima*, *Synedra acus*, and *Leptocylindrus danicus* exhibited a positive correlation with DO and NH_3_-N, while demonstrating a negative correlation with salinity and pH. Conversely, the abundance of *Paralia sulcata*, *Coscinodiscus asteromphalus*, *Tripos muelleri*, and *Coscinodiscus granii* was positively correlated with salinity and pH and negatively correlated with NH_3_-N, NO_2_-N, COD, and NO_3_-N.

## 4. Discussion

### 4.1. Plankton Species Composition and Seasonal Variation

The survey of Tahe Bay recorded 83 species of phytoplankton representing six phyla and forty-one genera; among them, 65 species of Bacillariophyta were the main taxa in this area, followed by 8 species of Pyrrophyta. The findings align with historical data from the central and southern Bohai Sea, as well as Bohai Bay, where Bacillariophyta maintained a significantly dominant presence in species count [8,31,32]. When *H’* < 1 and *J* < 0.3, then it can be determined that the diversity of phytoplankton communities is at a low level, with poor community stability and weak resistance to external disturbances [33]. In this study, the average value of H’ was higher than 1.5, and the value of J was greater than 0.3, so it can be judged that the state of phytoplankton community in this sea area is relatively stable, the number of species is relatively rich, and the ability to resist external disturbances is strong.

The abundance of phytoplankton in this marine region peaked in late winter, succeeded by spring, and decreased compared to autumn in late summer. The reasons for the higher abundance of phytoplankton in late winter and spring may be attributed to the self-purification capacity of the water body, which influences phytoplankton diversity and quantity [34]. Additionally, the relatively sluggish current and slow water exchange rate during winter may lead to a higher nutrient salt level of the water body, thereby affecting phytoplankton species and abundance. In spring, optimal water temperatures and nutrient-rich conditions create highly favorable conditions for phytoplankton proliferation and growth. This survey revealed that phytoplankton species and abundance in November 2022 (autumn) significantly exceeded those observed in late summer, contradicting the findings of Li et al. [35] regarding the phytoplankton community in the Bohai Sea, yet aligning with the results of Lin et al. [36] on phytoplankton community structure in the Weizhou Island sea area. The phytoplankton abundance in Tahe Bay waters is lower in late summer than in autumn for two primary reasons: Initially, water temperature is a crucial factor. The phytoplankton community in this region is predominantly composed of Bacillariophyta, which are suitable for growth in low-temperature and low-salt environments, and their abundance fluctuates markedly with the water temperature, with extreme values occurring around 18 °C and 21.5 °C [37,38]. However, the water temperature in Tahe Bay at the end of summer is approximately 23 °C, which is not conducive to the growth of diatoms. Furthermore, the marine fishery industry occupies an important position in Lushun, Dalian, with seawater aquaculture serving as a primary growth area, focusing on high-quality fish, shrimp, crab, and other species. In September, the hydrological conditions were conducive, the diversity of seawater cultured species was plentiful, and phytoplankton feeding intensified, particularly due to the high biomass of filter-feeding shellfish, which imposed feeding pressure on phytoplankton, leading to a relatively diminished phytoplankton abundance in late summer in the waters of Tahe Bay [39,40].

### 4.2. Ecological Niche Width Analysis

The niche width indicates the distribution of species and their capacity to utilize diverse environmental resources, with its size being related to the extent of species’ uptake of the environmental resources in which they are located and their degree of adaptation to the variable environment. The analysis of the niche width of dominant phytoplankton species reveals that, among the four sampling nodes, wide-niche species constituted 100% of the dominant species during late summer, winter, and spring, and there were six wide-niche species in autumn, which accounted for 66.6% of all dominant species of the season. This indicates that the dominant phytoplankton species in Tahe Bay are primarily wide-niche species, which are widely and uniformly distributed in the spatial dimension and have a strong ability to cope with environmental changes and resource utilization. The maximum niche width values were recorded in September 2021 for *Trieres chinensis* (1.00) and *Coscinodiscopsis jonesiana* (1.00), in March and April 2022 for *Thalassionema nitzschioides* (1.00), and in November 2022 for *Melosira granulata var. angustissima* (0.89).

In identical environmental conditions within the same marine region, a greater niche width signifies enhanced habitat suitability and resource appropriation by organisms, thereby increasing their competitive advantage in interspecific interactions, which implies that niche width serves as a critical metric for assessing the superior adaptive capacity and ecological superiority of phytoplankton within the biological community [41]. From the analysis of the dynamic change dimension of the number of phytoplankton dominant species, an obvious seasonal change rule was found. Specifically, the number of dominant species was marginally greater in spring and autumn compared to late winter and summer. Furthermore, the ecological niche width of these dominant species fluctuated with the seasons; for instance, *Thalassionema nitzschioides* and *Navicula cancellata* occupied wide ecological niches during late winter and spring; however, in autumn, they could not maintain their dominance in ecological competition and were gradually supplanted by *Bacillaria paradoxa Gmelin* and *Melosira granulata var. angustissima*. The ecological niche width of *Leptocylindrus danicus* in autumn was 0.70, which increased to 0.93 by the end of summer, and it became a wide ecological niche species with a larger ecological niche width at the end of the summer. To some extent, this phenomenon indicates that the niche width of species varies with the ecological environment, demonstrating that species exhibit heightened sensitivity to environmental alterations, resulting in corresponding changes in their adaptive capacity and resource utilization levels. He et al. [42] and Wei et al. [43] examined the ecological niches of phytoplankton in the eight estuaries of the Pearl River and Changhu Lake, respectively, concluding that a significant association exists between the niche width of dominant species and species abundance. In this study, species abundance was not a significant determinant of niche width; for instance, the lowest mean abundance of *Noctiluca scintillans* (325 species/m^3^) corresponded to a niche width of 0.95, whereas the mean abundance of *Chaetoceros densus* was 35.42 × 10^3^ species/m^3^, resulting in a niche width of only 0.69. This resembles the findings of Wang et al. [44] in their investigation of phytoplankton ecological niches in Zhanjiang. Therefore, the niche width of species reflects the degree of equilibrium and stability in the distribution and development process, which may be correlated with the coefficient of variation in the abundance of each species, while showing no significant correlation with the mean species abundance. Species exhibiting a higher *B_i_* value demonstrate broader distribution and more stable growth, and conversely, species with lower *B_i_* values exhibit narrower distribution and less stability.

### 4.3. Ecological Niche Overlap Analysis

The extent of niche overlap can evaluate the similarity of various species regarding resource usage and the possible competitive relationships that may arise from it [45]. The number of pairs of phytoplankton dominant species with significant niche overlap in the four sampling times of September 2021, March 2022, April 2022, and November 2022 in the waters of Tahe Bay accounted for 40.0%, 100%, 93%, and 58% of the total number of pairs, respectively. This indicates that there are commonalities in resource use across species and that species competition for similar resources is related to resource availability. Insufficient resource supply will markedly elevate the intensity of competition among species [46,47]. RDA and CCA, as the fundamental methodologies for elucidating the link between species and environmental variables, were examined to demonstrate that the magnitude of niche overlap values among pairs of species was significantly associated with their correlation to environmental factors. Species pairs with low niche overlap values were mainly characterized by two types of situations: firstly, species pairs with significant negative correlation with the same environmental factor; secondly, species pairs with significant correlation with different environmental factors. For example, the species pairs of *Leptocylindrus danicus* and *Trieres chinensis*, as well as *Coscinodiscopsis jonesiana*, and the species pairs of *Noctiluca scintillans* and *Trieres chinensis*, together with *Coscinodiscopsis jonesiana*. Likewise, if the species pairs exhibit a substantial positive correlation with the same environmental factor, their ecological niche overlap values are relatively high, as shown in *Trieres chinensis*—*Coscinodiscopsis jonesiana* and *Thalassiosira nordenskioeldii*—*Chaetoceros densus*, among others. In the existing findings, those dominant species with larger values of niche widths usually also showed more significant niche overlap; however, certain species with high niche widths values had a lower degree of overlap but accounted for a lower proportion of the total, which was similar to the results of the study by Gong et al. [48] in Dongping Lake. For example, the ecological niche overlap index between *Leptocylindrus danicus* (0.94 for *B_i_*) and *Trieres chinensis* (1.00 for *B_i_*) was 0. It might be due to the high degree of spatial ecological niche differentiation in these two dominant phytoplankton species, which showed high heterogeneity in environmental resources, as well as the ecological niche characteristics of different species and differences in biological traits. Therefore, when evaluating the competitive dynamics between species, the results obtained only by the single index of niche width are likely to have some deviation from the real situation [49].

### 4.4. Analysis of Ecological Niche Differentiation of Dominant Phytoplankton Species About Environmental Factors

This investigation identified a total of 20 dominant phytoplankton species across three phyla, and the dominant phytoplankton species were dominated by Bacillariophyta. The highest number of dominant species was observed in autumn (nine species), followed by spring (eight species), late winter (seven species), and late summer (five species). It has been shown that the number of diatom populations is positively correlated with nitrogen and phosphorus nutrients [50,51], and diatoms exhibit considerable adaptability to water temperature and light. Consequently, one of the reasons for the largest percentage of diatoms may be that the study waters are relatively rich in nitrogen and phosphorus nutrients. Throughout the examination period, the water temperature ranged from 5 to 25 °C, while salinity varied between 30.2 and 32.2, which made the environment of this area relatively stable. Therefore, the composition of phytoplankton species in this area did not change much in all seasons. Water temperature, salinity, and nutrient salts are the primary environmental factors influencing phytoplankton [52,53,54]. Diatoms adapted to low temperatures, and salinity emerged as the dominant species in the low salinity of Tahe Bay during autumn. Additionally, Cyanobacteria with nitrogen-fixing ability, such as *Bacillaria paradoxa Gmelin*, and methanobacteria with higher nutrient uptake ability, like *Tripos muelleri*, were able to grow and become the dominant species [34], thereby enhancing the diversity and composition of dominant phytoplankton species in the autumn. Moreover, other factors, including water temperature, waves, and the eutrophication condition of the water body, may be related to the seasonal changes in dominant species. The offshore waters of Dalian have strong winds in winter, and there is a significant positive correlation between wave impact power, wind size, and the abundance of *Paralia sulcata*, whose dominance was extremely pronounced during this season.

Pearson’s correlation analysis revealed a strong correlation between phytoplankton diversity indices (*H’*, *D*, *J*) and several environmental factors, including water temperature, salinity, pH, and DO. Consequently, these environmental factors may affect the composition of the phytoplankton community by influencing the phytoplankton diversity indices (H’, D, J). Different nutrient salt structure ratios may influence the abundance of Pyrrophyta, Chrysophyta, and Cyanophyta, as well as diversity indices and evenness; for instance, an increase in nitrate and inorganic phosphorus content, to some extent, may inhibit the growth of these organisms. The integrated findings of CCA and RDA indicated that the primary environmental variables influencing the ecological differentiation of dominant phytoplankton species are water temperature, DIP, COD, NO_3_-N, NO_2_-N, and NH_3_-N, corroborating the results of Yang et al. [55] regarding seasonal variations in phytoplankton in Liaodong Bay. Water temperature, a critical determinant of biological enzyme activity in phytoplankton metabolism, can directly influence phytoplankton abundance and community structure alterations through its metabolic effects, with varying responses among different phytoplankton species [56,57]. In this study, several higher abundance species, such as the second dominant species, *Thalassiothrix longissima*, had a significant positive correlation with water temperature and a significant negative correlation with COD (Figure 3). The greater prevalence of *Chaetoceros peruvianus* in spring compared to winter may be ascribed to the positive correlation between water temperature and *Chaetoceros peruvianus* during spring, and the increase in water temperature would promote their reproduction. This suggests that the reduced water temperature likely serves as the primary constraint on the growth of the dominant species, *Chaetoceros peruvianus*, in winter, and the RDA findings of this study indicated a negative association between the dominant species in winter and water temperature (Figure 4). When phytoplankton carry out photosynthesis, carbon dioxide in seawater is consumed, resulting in an elevation in pH; however, when photosynthesis intensity reaches a certain level, elevated pH levels will inhibit the process of photosynthesis. Consequently, pH is not only a key element in determining whether phytoplankton photosynthesis can be carried out efficiently but also an effect triggered by photosynthesis [1]. Nutrient salts are essential for the growth and reproduction of phytoplankton, and their concentrations play an important role in phytoplankton growth [58,59]. In this investigation, Pearson’s correlation analysis revealed that NO_2_-N showed a significant negative correlation with the abundance of Pyrrophyta and Cyanophyta, as well as a very significant negative correlation with the abundance of Chrysophyta. The majority of the dominant species in the four sampling sites were positively correlated with nitrogen and phosphorus nutrients, which is consistent with the results of Chen et al. [12] but differed from those obtained by Li et al. [60]. This discrepancy may be attributed to the nutrient composition of the respective study areas, the dominant phytoplankton species, and the water temperature during sampling.

### 4.5. Analysis of the Possible Variability of Phytoplankton Diversity

The results of the study basically represent the seasonal changes in biodiversity in the study waters. In this study, a total of 16 stations were set up in the study area, and phytoplankton samples were collected, stored, and transported in accordance with the [16], and the identification of sample species and quantitative analysis were performed by a qualified organization accredited by CMA (China Inspection Body and Laboratory Mandatory), so the data were representative.

In recent years, the climate has undergone significant changes, and this environmental change may affect phytoplankton’s primary productivity and community structure. When Boris et al. [61] quantified the resilience of phytoplankton communities to environmental stresses through adaptive evolution, revealing that the current Earth system model, which presumes that phytoplankton are fully adapted to temperature, may overestimate their resilience to climate change; Henson et al. [62] analyzed a complex ecosystem model comprising 35 phytoplankton species to evaluate alterations in phytoplankton community composition, turnover, and size structure in the 21st century, discovering that the turnover of phytoplankton communities has become faster in this century, and that community structure has become increasingly unstable under the influence of climate change; Gittings et al. [63] identified that future climate warming may affect phytoplankton growth in tropical marine ecosystems in two ways: (1) reduction in phytoplankton abundance and (2) a shift in seasonal phytoplankton blooming times. Consequently, phytoplankton biodiversity in this study area may also change due to climate change, resulting in changes in community structure and turnover of dominant species. This requires further related investigations and studies over a longer time span to reveal the relationship between the response of biological communities and the climate change.

## 5. Conclusions

During the investigation in the waters of Tahe Bay, 83 species of phytoplankton were identified in six phyla and 41 genera, dominated by Bacillariophyta (65 species, accounting for 78.31%); the abundance of phytoplankton varied from 14.90 × 10^3^ to 17.46 × 10^5^ ind·m^−3^, the highest at the end of winter and the lowest at the end of summer; the phytoplankton community in this sea area has a high diversity of species, and the ability of resisting external disturbances is relatively strong.

A total of 20 dominant phytoplankton species were identified in three phyla, with niche widths ranging from 0.44 to 1.00, and mainly consisted of wide-niche species. The extent of niche overlap among the dominant species varied seasonally, with severe overlap proportions of 58.0% in autumn, 40.0% in summer, 93.0% in spring, and 100% in winter. The competition among phytoplankton species appears to be most intense in winter, followed by spring and autumn, and lowest in summer.

The abundance and community diversity of phytoplankton in Tahe Bay were significantly correlated with environmental variables, including water temperature, salinity, pH, DO, NO_3_-N, and DIP. The increase in NO_3_-N and DIP would probably inhibit the growth of species in the Pyrrophyta, Chrysophyta, and Cyanophyta. The primary environmental parameters influencing the biological niche differentiation of dominant phytoplankton species were COD, water temperature, NO_3_-N, DIP, NO_2_-N, and NH_3_-N.

Global climate change has become increasingly significant in recent years, and this environmental change may affect the structure of the plankton community and the turnover of dominant species. Meaningful analysis of these changes requires long-term survey data. Therefore, this study will continue to carry out relevant surveys to accumulate data for analyzing the response of plankton to climate change.

## Figures and Tables

**Figure 1 biology-14-00578-f001:**
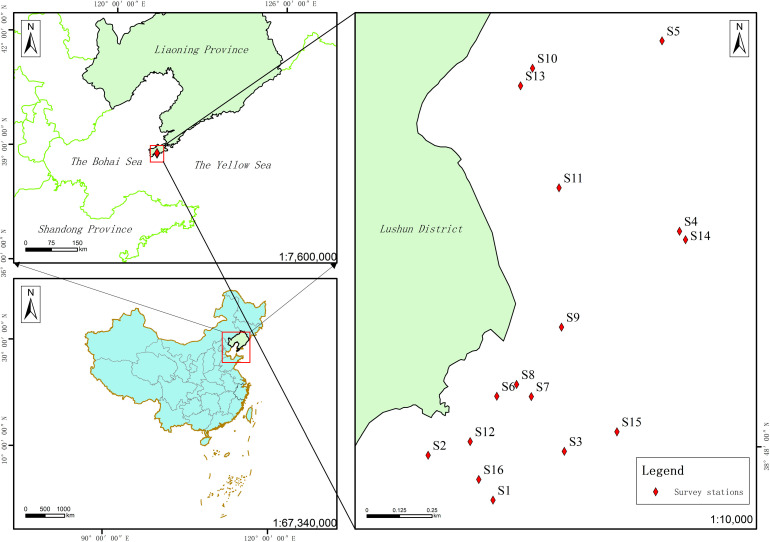
Map of survey stations in the maritime region of Tahe Bay, Lushun.

**Figure 2 biology-14-00578-f002:**
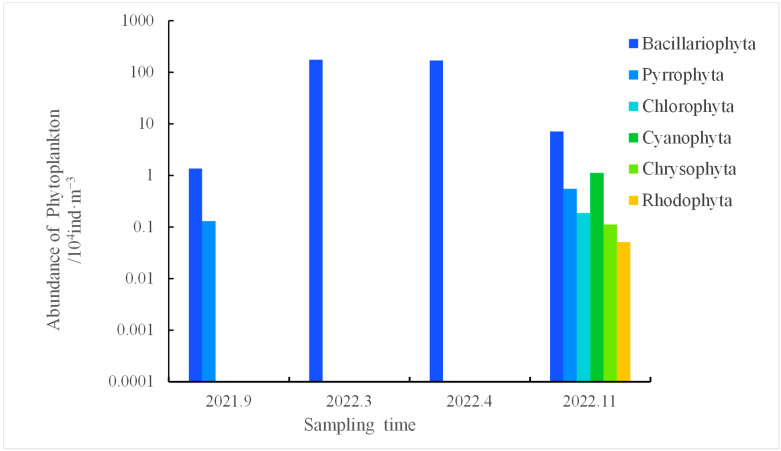
The abundance of phytoplankton at various sample intervals in the waters of Tahe Bay.

**Figure 3 biology-14-00578-f003:**
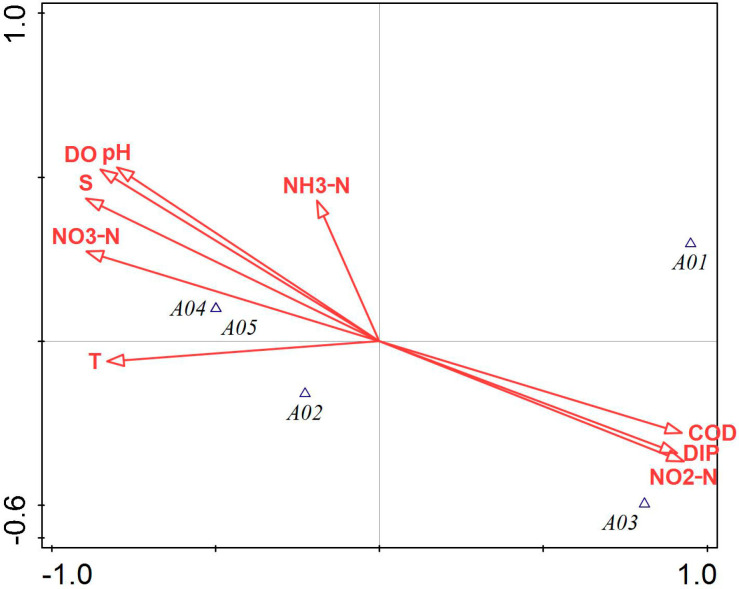
Canonical correspondence analysis of phytoplankton and environmental factors in September 2021. Note: The first principal component of the graph contributes 81.78%, and the second principal component contributes 12.39% of the sample variation. A01: *Leptocylindrus danicus*; A02: *Thalassiothrix longissima*; A03: *Noctiluca scintillans*; A04: *Trieres chinensis*; A05: *Coscinodiscopsis jonesiana*; A06: *Paralia sulcata*; A07: *Thalassiosira nordenskioeldii*.

**Figure 4 biology-14-00578-f004:**
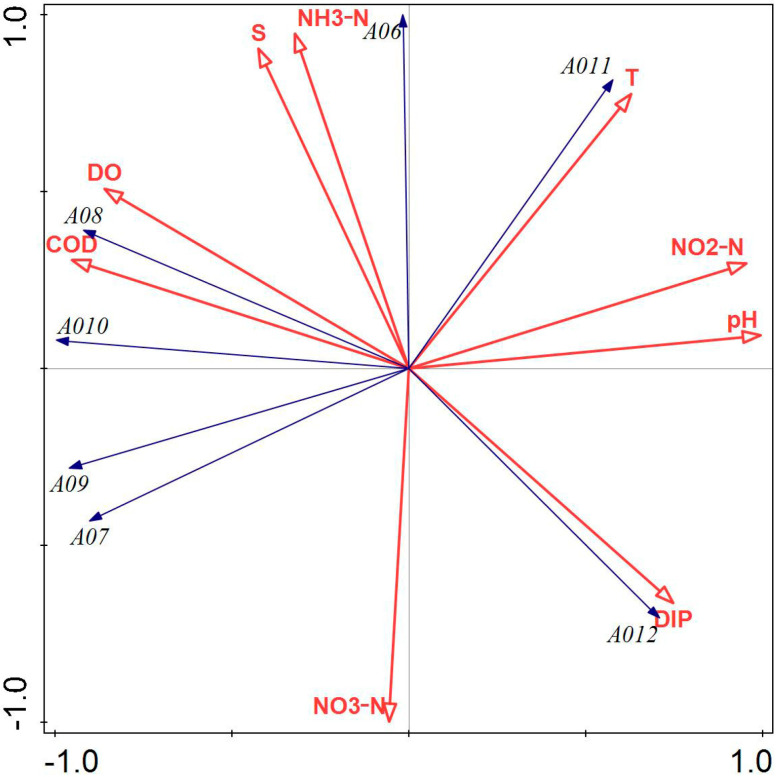
Redundancy analysis of phytoplankton and environmental factors in March 2022. Note: The first principal component of the graph contributes 81.69%, and the second principal component contributes 18.31% to sample variation. A06: *Paralia sulcata;* A07: *Thalassiosira nordenskioeldii*; A08: *Stephanopyxis turris*; A09: *Chaetoceros densus*; A010: *Chaetoceros peruvianus*; A011: *Thalassionema nitzschioides*; A012: *Navicula cancellata*.

**Figure 5 biology-14-00578-f005:**
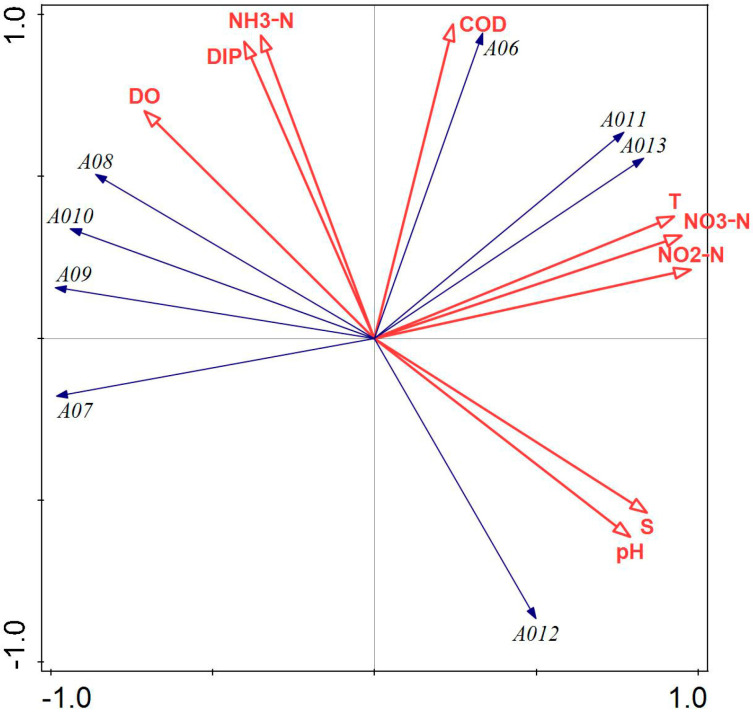
Redundancy analysis of phytoplankton and environmental factors in April 2022. Note: The first principal component of the plot contributes 72.10%, and the second principal component contributes 27.90% to sample differences. A06: *Paralia sulcata;* A07: *Thalassiosira nordenskioeldii*; A08: *Stephanopyxis turris*; A09: *Chaetoceros densus*; A010: *Chaetoceros peruvianus*; A011: *Thalassionema nitzschioides*; A012: *Navicula cancellata;* A013: *Skeletonema costatum*.

**Figure 6 biology-14-00578-f006:**
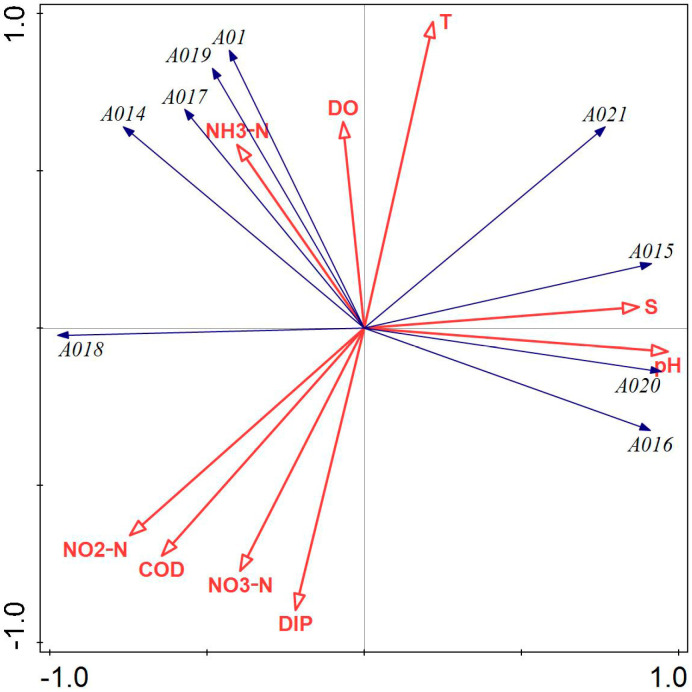
Redundancy analysis of phytoplankton and environmental factors in November 2022. Note: The first principal component of the graph contributes 58.28%, and the second principal component contributes 39.58% to the sample variation. A01: *Leptocylindrus danicus*; A014: *Bacillaria paradoxa Gmelin*; A015: *Coscinodiscus asteromphalus*; A016: *Coscinodiscus granii*; A017: *Licmophora abbreviata*; A018: *Melosira granulata var. angustissima*; A019: *Synedra acus*; A020: *Tripos muelleri*; A06: *Paralia sulcata*.

**Table 1 biology-14-00578-t001:** Number of phytoplankton species in the waters of Tahe Bay in 4 different seasons.

Taxa	September 2021 (End of Summer)	March 2022 (End of Winter)	April 2022 (Spring)	November 2022 (Autumn)
Bacillariophyta	8	36	27	41
Cyanophyta	1	0	0	7
Pyrrophyta	0	0	0	3
Chlorophyta	0	0	0	1
Chrysophyta	0	0	0	4
Rhodophyta	0	0	0	2

**Table 2 biology-14-00578-t002:** Number of phytoplankton species and assessment indices in the marine region of Tahe Bay.

Sampling Time	Phytoplankton			
	Number of Species (S)	Shannon–Wiener’s Diversity Index (*H’*)	Margalef’s Index (*D*)	Pielou Index (*J*)
September 2021	9	2.03	0.83	0.84
March 2022	36	2.42	2.44	0.31
April 2022	27	2.22	1.81	0.34
November 2022	58	2.80	4.99	0.28
Mean	32.5 ± 20.4	2.37 ± 0.33	2.52 ± 1.78	0.45 ± 0.26
Median	31.5	2.32	2.13	0.33

**Table 3 biology-14-00578-t003:** Dominant phytoplankton species in the waters of Tahe Bay.

Serial Number	Species Name	Dominance Index			
		September 2021 (End of Summer)	March 2022 (End of Winter)	April 2022 (Spring)	November 2022 (Autumn)
A01	*Leptocylindrus danicus*	0.116			0.147
A02	*Thalassiothrix longissima*	0.111			
A03	*Noctiluca scintillans*	0.044			
A04	*Trieres chinensis*	0.042			
A05	*Coscinodiscopsis jonesiana*	0.042			
A06	*Paralia sulcata*		0.364	0.391	0.165
A07	*Thalassiosira nordenskioeldii*		0.045	0.047	
A08	*Stephanopyxis turris*		0.041	0.026	
A09	*Chaetoceros densus*		0.061	0.070	
A010	*Chaetoceros peruvianus*		0.102	0.106	
A011	*Thalassionema nitzschioides*		0.111	0.115	
A012	*Navicula cancellata*		0.063	0.065	
A013	*Skeletonema costatum*			0.020	
A014	*Bacillaria paradoxa Gmelin*				0.062
A015	*Coscinodiscus asteromphalus*				0.021
A016	*Coscinodiscus granii*				0.027
A017	*Licmophora abbreviata*				0.047
A018	*Melosira granulata var. angustissima*				0.148
A019	*Synedra acus*				0.029
A020	*Tripos muelleri*				0.038

**Table 4 biology-14-00578-t004:** Phytoplankton niche width (*B_i_*) and niche overlap (*Q_ik_*) statistics in September 2021.

Code	*B* * _i_ *	*Q* * _i_ * * _k_ *				
		A01	A02	A03	A04	A05
A01	0.93	1				
A02	0.79	0.18	1			
A03	0.95	0.88	0.26	1		
A04	1.00	0.00	0.95	0.00	1	
A05	1.00	0.00	0.95	0.00	1.00	1

Note: See Table 3 for species codes in the figure.

**Table 5 biology-14-00578-t005:** Phytoplankton niche width (*B_i_*) and niche overlap (*Q_ik_*) statistics for March 2022.

Code	*B* * _i_ *	*Q* * _i_ * * _k_ *						
		A06	A07	A08	A09	A010	A011	A012
A06	0.87	1.00						
A07	0.77	0.74	1.00					
A08	0.77	0.92	0.86	1.00				
A09	0.69	0.70	1.00	0.85	1.00			
A010	0.86	0.88	0.96	0.96	0.95	1.00		
A011	1.00	0.95	0.85	0.87	0.81	0.91	1.00	
A012	0.97	0.87	0.84	0.78	0.79	0.86	0.98	1.00

Note: See Table 3 for species codes in the figure.

**Table 6 biology-14-00578-t006:** Phytoplankton niche width (*B_i_*) and niche overlap (*Q_ik_*) statistics for April 2022.

Code	*B* * _i_ *		*Q* * _i_ * * _k_ *						
		A06	A07	A013	A08	A09	A010	A011	A012
A06	0.85	1.00							
A07	0.77	0.71	1.00						
A013	0.93	0.95	0.45	1.00					
A08	0.98	0.86	0.84	0.70	1.00				
A09	0.74	0.73	0.99	0.48	0.89	1.00			
A010	0.86	0.86	0.96	0.65	0.94	0.98	1.00		
A011	1.00	0.94	0.85	0.82	0.80	0.84	0.91	1.00	
A012	0.97	0.86	0.84	0.73	0.69	0.80	0.86	0.98	1.00

Note: See Table 3 for species codes in the figure.

**Table 7 biology-14-00578-t007:** Phytoplankton niche width (*B_i_*) and niche overlap (*Q_ik_*) statistics in November 2022.

Code	*B* * _i_ *	*Q* * _i_ * * _k_ *								
		A014	A015	A016	A01	A017	A018	A019	A020	A06
A014	0.79	1.00								
A015	0.65	0.41	1.00							
A016	0.44	0.17	0.89	1.00						
A01	0.70	0.84	0.68	0.32	1.00					
A017	0.52	0.64	0.65	0.26	0.95	1.00				
A018	0.89	0.74	0.61	0.45	0.81	0.72	1.00			
A019	0.67	0.78	0.68	0.31	1.00	0.98	0.80	1.00		
A020	0.59	0.38	0.89	0.97	0.44	0.31	0.55	0.41	1.00	
A06	0.61	0.55	0.79	0.43	0.90	0.94	0.57	0.92	0.46	1.00

Note: See Table 3 for species codes in the figure.

**Table 8 biology-14-00578-t008:** Correlation of phytoplankton community indicators with environmental factors.

Phytoplankton Community Indicator	Water Temperature (°C)	Salinity	pH	Dissolved Oxygen Concentration (mg/L)	Chemical Oxygen Demand (mg/L)	Ammonia–Nitrogen Content (μg/L)	Nitrate–Nitrogen Content (μg/L)	Nitrite–Nitrogen Content (μg/L)	Inorganic Phosphorus Content (μg/L)
*H′*	−0.852 **	0.795 **	−0.39	0.528	0.058	−0.888 **	−0.797 **	−0.792 **	−0.083
*J*	0.939 **	−0.884 **	0.321	−0.633 *	−0.288	0.938 **	0.770 **	0.872 **	−0.054
*D*	−0.901 **	0.929 **	−0.756 **	0.247	0.16	−0.809 **	−0.928 **	−0.788 **	−0.443
Abundance of Bacillariophyta	−0.354	0.131	0.593 *	0.960 **	0.419	−0.404	0.043	−0.357	0.765 **
Abundance of Pyrrophyta	−0.331	0.518	−0.829 **	−0.418	−0.095	−0.287	−0.550 *	−0.348	−0.699 **
Abundance of Rhodophyta	−0.271	0.401	−0.478	−0.068	−0.206	−0.227	−0.401	−0.232	−0.385
Abundance of Chlorophyta	−0.316	0.32	−0.513	−0.15	0.296	−0.236	−0.416	−0.241	−0.399
Abundance of Chrysophyta	−0.463	0.581 *	−0.800 **	−0.19	−0.042	−0.375	−0.661 **	−0.383	−0.634 *
Abundance of Cyanophyta	−0.407	0.535 *	−0.715 **	−0.161	−0.141	−0.336	−0.593 *	−0.343	−0.569 *

Note: “*” indicates a significant correlation at the 0.05 level (two-sided); “**” indicates a highly significant correlation at the 0.01 level (two-sided).

## Data Availability

Data are contained within the article.

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
