# Peer review of "Phytoplankton Structure and Ecological Niche Differentiation of Dominant Species in Tahe Bay, China"

_biology, 2025, doi:10.3390/biology14050578_

Round 1

Reviewer 1 Report

Comments and Suggestions for Authors

This manuscript is thoroughly analysed, described, and discussed based on datasets from four seasons of the year. I have provided comments on the manuscript, and I hope they will be useful for the revision. As I noted at the end, since this study was conducted over just one year, you may want to especially discuss the temporal variation in phytoplankton diversity.

Simple Summary:

The description of the last paragraph, “The results of the study provide scientific references for local high-quality marine-enhanced aquaculture production,” seems weak as a summary. You might focus more on your findings regarding the high diversity at the study site, which should contribute to the sustainability of the local ecosystem, not only for aquaculture, among other areas.

Abstract:

It should also include one or two summarized sentences to highlight the novelty and importance of this research.

Introduction:

2nd and 3rd paragraphs. Since this journal has an international focus, please explain why you specifically conducted your study in the waters of Dalian or Dayaowan Bay, in addition to addressing their regional interest and significance.

The last sentence, “This will elucidate the alteration rules of dominant species and the stability of the community structure of the phytoplankton community in this sea area, thereby offering a theoretical foundation for the scientific assessment of the ecological health of the area and the sustainable utilization of marine biological resources.” is relevant to state in the Abstract, Discussion, or both.

  1. Materials and Methods

2.3. Measurement of Physical and Chemical Indicators of Water Quality

Since you measured the COD, why didn’t you measure the BOD? Generally, T, salinity, pH, and dissolved oxygen are sufficient as relevant parameters for the research. The authors may explicitly justify the reason for measuring COD.

2.4.5. Mathematical Statistics

Mathematical Statistics” should be “Statistics”.

  1. Results and Analyses

Check significant digits thoroughly throughout the results. For example, L202, “ranged from 1.49 × 104 to 17.4625 × 105ind·m−3.” These values are inconsistent with each other, and there are many such inconsistencies throughout the manuscript.

  1. Discussion

4.1. Plankton Species Composition and Seasonal Variation

If the authors provide the detailed number of species recorded in previous studies, it would assist readers in understanding the historical changes in plankton diversity in the area.

The discussion is well summarised. However, this research was conducted over one year, so we do not know whether the outcomes are representative of the study site or if there might be potential annual spatial and temporal variations. You may discuss the possible variability of plankton diversity. Conversely, if this research provides representative data, you may justify it in the discussion. Since climate change has varied significantly in recent years, such environmental variability could impact the plankton community. You may also highlight this issue in the conclusion section.

Comments on the Quality of English Language

I suggest the authors ask for proofreading by a native English speaker.

Reviewer 2 Report

Comments and Suggestions for Authors

The article by Zhang et al. is in the field of research of a very important and interesting problem – differentiation of phytoplankton species by ecological niches and refers us to understanding the “Phytoplankton Paradox”. There are few works in this direction, and the authors close the gap of lack of knowledge. Using the example of Tahe Bay, China, the authors set the goal of studying the taxonomic structure of phytoplankton depending on the parameters of the habitat. They determined which environmental factors are key in the distribution of dominant phyla. As such, the hypothesis is not formulated in the Introduction, but, as I understand it, the authors had in mind the hypothesis that the taxonomic structure of phytoplankton depends on the parameters of the habitat. It was previously known (e.g. Habib et al. Hydrobiologia, 1997, 350, 63-79; Resende et al. Acta Oceanologica, 2007, 32, 224-235; Zhu et al. Environ. Monit. China, 2024, 40, 129-142), that phytoplankton dynamics depend on environmental factors. The authors added to this body of knowledge with their study. In addition to the fundamental significance, the results obtained are also important for solving practical issues of marine aquaculture optimization. The title of the article reflects the essence of the work. The Summary is written according to the canons. The keywords repeat the words from the title of the article, it is better to avoid this to increase the relevance of the article.The methods are described in detail, necessary references and formulas are given, which allows to check or reproduce similar studies. I have no suggestions for improving the research methodology. Results and Discussion are written in separate chapters and divided into sub-chapters. The obtained data are subjected to statistical processing. The article contains 6 Figures and 8 tables. The information in them is presented clearly, the captions are informative. The Conclusion is based on the results obtained.

Some wishes for illustrations. In Figure 1 (Map), it is desirable to show on the globe where the research site is located. In the tables, it is desirable to provide measurement errors.

The references are appropriate. The list of reviewed literature contains 48% of references to publications of the last 5 years, 12% are self-citations.

Round 2

Reviewer 1 Report

Comments and Suggestions for Authors

Thank you for your thorough revision and detailed responses to my comments. I am delighted with your revision.